# Effect of Lifting Gas Diffusion on the Station-Keeping Performance of a Near-Space Aerostat

**Jun Li, Linyu Ling, Jun Liao \*, Zheng Chen and Shibin Luo**

School of Aeronautics and Astronautics, Central South University, Changsha 410083, China; lijun215@csu.edu.cn (J.L.); 15797890218@163.com (L.L.); chenzheng@csu.edu.cn (Z.C.); luoshibin@csu.edu.cn (S.L.)

\* Correspondence: liaojun@csu.edu.cn

**Abstract:** During the long-endurance flight of a near-space aerostat, the characteristics of lifting gas diffusion have a great influence on the flight altitude adjustment and station-keeping performance. Thus, in this study, a lifting gas diffusion model and a dynamic model that consider thermal effects, which had not been studied in similar models before, were developed. The dynamic model and thermal model were validated by historic flight data, and the calculated lifting gas diffusion results were compared with the experimental data of other researchers. The variations in the flight endurance, flight altitude, lifting gas diffusion rate, and diffusion coefficient of a near-space aerostat were analyzed. The effects of the ratio of porosity to tortuosity and envelope radiation properties on the mass of the lifting gas and flight altitude were considered in detail. To analyze the effect mechanism of the ratio of porosity to tortuosity and the envelope radiation properties, the envelope and gas temperature, as well as the gas pressure, were studied. The results show that the lifting gas diffusion rate and diffusion coefficient are very sensitive to the change in the ratio of porosity to tortuosity and envelope temperature. The results obtained from the analysis of the lifting gas diffusion can lay a solid foundation for improving the flight performance of near-space aerostats and for providing improved design considerations for aerostats.

**Keywords:** lifting gas diffusion; near-space aerostat; theoretical study; thermal effect; station-keeping performance

## 1. Introduction

In recent years, more and more scientific research institutions and scholars have focused on utilizing lighter-than-air aerial flight vehicles as pseudo-satellites operating for extended durations of time in near space to achieve relevant military and civilian tasks previously accomplished using traditional spacecraft [1]. A near-space aerostat is an ideal high-altitude platform for Earth observation, homeland security, data and communications relay, and the relay of communications that need the regional station-keeping ability to adapt to multiple environments [2]. Whether in national defense or economic activities, a near-space aerostat can be used as a supplement to "space-based" vehicles.

For a near-space aerostat, its station-keeping performance is its main technical advantage [3]. To address station-keeping performance requirements, many scholars and researchers have carried out theoretical studies and simulations [4]. Ramesh et al. [5] evaluated the stationary performance of a dual-balloon system and a balloon–stratosail system and compared the best ranges of wind conditions for tether systems. The results showed that the dual-balloon system is more suitable for passive control in a wind field with quasi-zero wind layer and that a balloon–stratosail system can carry out appropriate wind resistance control. Van Wynsberghe et al. [6] presented an innovative design for a stratospheric platform that can achieve long-endurance horizontal station-keeping using electrohydrodynamic thrusters. Another innovation of this paper was its presentation of a wireless energy supplement scheme. Yang et al. [7] proposed a new conceptual design

method and fuzzy adaptive backstepping control approach to adapt to inaccurate and non-real-time high-altitude wind field data. Based on the phenomenon of a quasi-zero wind layer at the bottom of the stratosphere, Du et al. [8] proposed the flight altitude adjustment method based on charging and discharging air in an aerostat ballonet to realize regional stationary flight or fixed-point air parking.

These studies provided a theoretical base for designing and controlling a near-space aerostat. These studies were mainly focused on flight performance, station-keeping endurance, and float altitude control. However, discussions of the effect of lifting gas diffusion on the station-keeping performance are rare. We all know that lifting gas diffusion is one of key elements in the design of envelope materials. Hall and Yavrouian [9] analyzed the effect of lifting gas loss on the internal gas pressure of a Venus super-pressure balloon (see in Figure 1). As a typical super-pressure lighter-than-air vehicle, the balloon's net buoyancy and maximum flight altitude will gradually reduce with the loss of lifting gas. The loss of flight altitude adjustment capability means that there will be a reduction in maneuverability. During flight, the supplementation of lifting gas is almost impossible for high-altitude aerostats. Therefore, the lifting gas diffusion directly determines the success or failure of long-endurance missions carried out by near-space aerostats. The thermal performance and leakage of lifting gas have a direct impact on the flight altitude control of aerostats. The question of how to fully understand the temperature, pressure, and diffusion of lifting gas is one of the most important issues needing to be addressed in the process of the real-time flight control of an aerostat.

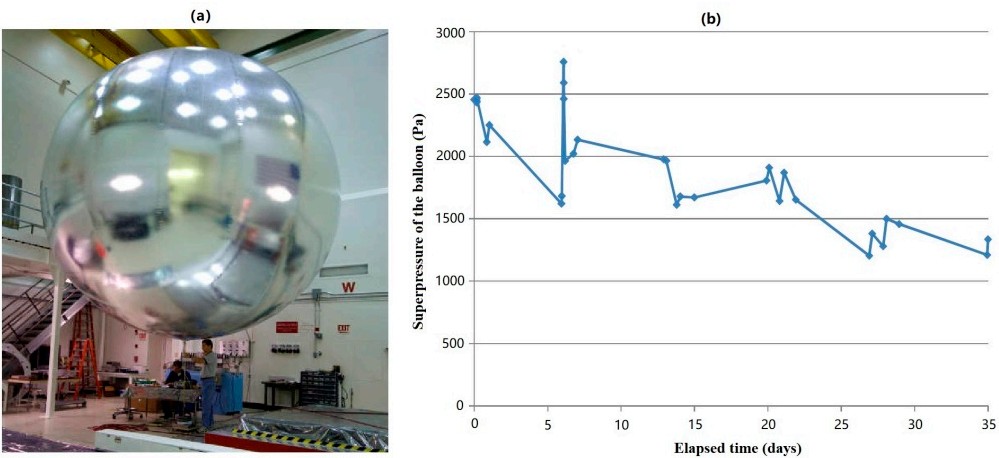

**Figure 1.** (**a**) A 5.5 m diameter prototype balloon undergoing testing; (**b**) 35-day buoyancy test data [9].

Inspired by studies of gas diffusion in the microelliptical groove gas diffusion layer, porous gas diffusion layer, and fibrous materials [10] under the influence of lifting gas loss, the gas diffusion characteristics of the envelope were studied here. The envelope is one of the major structural parts of a near-space aerostat. It is used to maintain the aerodynamic shape of an aerostat by filling it with lifting gas. The envelope material of a high-altitude aerostat is composed of a laminated membrane [11]. An envelope is made of multi-layer materials, including wearable, ultraviolet, lifting gas barrier, UV protection, anti-aging, and load-bearing layers, as shown in Figure 2 [12].

To date, the study of lifting gas diffusion caused by composite damage and external load has been largely focused on the mechanical properties of carbon-fiber-reinforced composites and the mode of crack propagation [13]. In one study, the lifting gas diffusion of orthogonal carbon-fiber-reinforced composite laminates under biaxial loading was measured [14]. The experimental results showed that the lifting gas diffusion coefficient on the surface of the laminates has a clear relationship with the load. In order to analyze the influence of thermodynamic properties and material matrix cracks, a diffusion model was developed obtain the diffusion coefficients of the composite laminates. It was found that

the smaller the ply angle of the material matrix was (the reference values were 45°, 60°, 75° and 90°), the larger leak conductance at the crack intersection would be [15]. Based on the Lennard–Jones interactions, Thornton et al. [16] proposed a transport model of gas in various media to predict the minimum pore size needed for Knudsen diffusion. Yao et al. [17] studied the lifting gas leakage mechanism of damaged flexible composite materials and analyzed the influence of different types of damage on lifting gas leakage; this was of significance for the study of the damage-induced lifting gas leakage of envelope materials. The structure of an envelope material is different from that of carbon-fiber-reinforced composite laminates, and the mechanism of lifting gas diffusion caused by damage is different [18]. To study the lifting gas diffusion of an envelope material, it is necessary to establish a corresponding theoretical model according to its structural characteristics and use new experimental methods to obtain more accurate results.

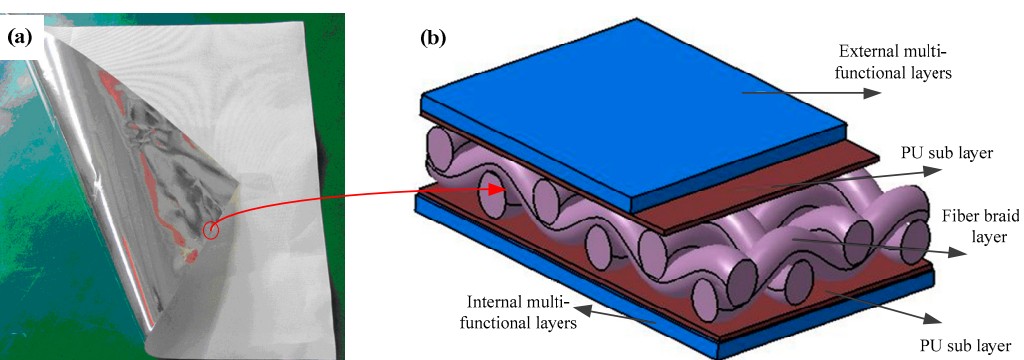

**Figure 2.** (**a**) Photograph of envelope; (**b**) cross-section of the aerostat envelope [12].

In view of the lightweight envelope material used and the harsh high-altitude thermal environment, the envelope of an aerostat is required to be processed very thinly, and the multi-functional composite layer has different mechanical properties that will lead to delamination and the damage of the envelope after being subjected to alternating thermal stress. Therefore, the study of the lift gas diffusion characteristics of flexible envelope materials is important for improving the station-keeping endurance and safety of near-space aerostats [19]. Based on a literature search, it can be seen that the pressure difference between the internal lifting gas and the external atmosphere can directly affect lifting gas diffusion at high altitudes. The gas pressure difference is directly related to the thermal performance of the lifting gas and the envelope [20]. Many experts and scholars have studied the thermal performance of aerostats [21]. Xiong et al. [22] proposed a simplified thermal model suitable for engineering to assess the steady equilibrium temperature in aerostats at float altitude. In addition to the temperature characteristics of the lifting gas and envelope, the thermal performance of the electronics within the system also needs to be studied to optimize the used thermal control methods.

From the abovementioned literature review, it can be seen that most studies to date have mainly focused on several key issues, such as the flight performance, station-keeping endurance, and thermal performance of near-space aerostats and the gas diffusion of flexible composite materials. However, investigations of the effects of the lifting gas diffusion on the station-keeping performance of a near-space aerostat have been rare. During the long-range flight of a near-space aerostat, it is necessary to understand the lifting gas diffusion characteristics. In this study of a naturally shaped super-pressure balloon made of laminated composite materials (as shown in Figure 2b), a lifting gas diffusion model and a dynamic model that considered thermal effects were developed. A numerical model was used to analyze the lifting gas diffusion and to study the effects of the envelope properties on the station-keeping endurance of the aerostat in detail. The contributions of this paper are summarized as follows: (1) A lifting gas diffusion model that considers thermal effects was established. The temperatures of the envelope and lifting gas were taken into consideration in this paper for the first time. The influence of the

envelope thickness variation was considered in the theoretical model to make the lifting gas diffusion model more precise. (2) The effects of the main air tightness parameters of the envelope and the envelope radiation properties on the lifting gas diffusion coefficient were analyzed. The relationship between gas diffusion and flight duration was analyzed. The relationship between lifting gas diffusion and flight endurance and the mechanism of the variation of the gas diffusion coefficient were studied. The results obtained from the analysis of the lifting gas diffusion can contribute to improving the flight performance and station-keeping endurance of near-space aerostats and to providing improved design considerations for aerostats.

## 2. Theoretical Model

### 2.1. Lifting Gas Diffusion Model

In near-space aerostats, the lifting gas diffuses out by passing through the envelope material. Figure 3 represents the mass transfer mechanism of lifting gas through an aerostat envelope.

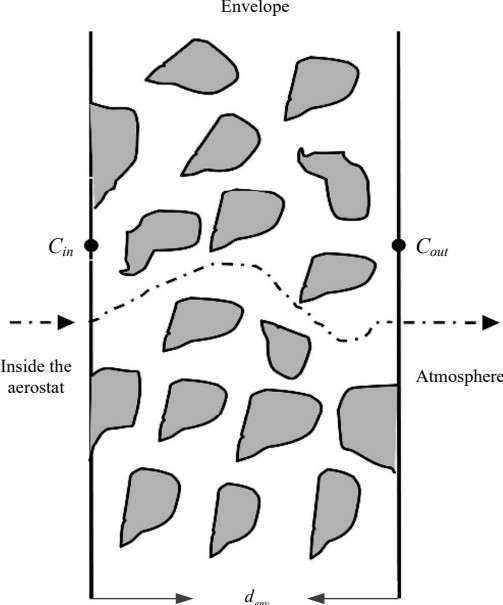

**Figure 3.** The mass transfer mechanism of lifting gas passing through an envelope [23].

Considering Figure 3, which demonstrates a case with a small envelope thicknesses, the gross mass of the diffusion lifting gas can be calculated as:

$$m_{diffusion} = M_{gas} \cdot n_{diffusion} \tag{1}$$

where $M_{gas}$ is the lifting gas molecular weight in kg/kmol.

$n_{diffusion}$ is the amount of lifting gas that diffuses. The differential amount of lifting gas that diffuses can be calculated as:

$$dn_{diffusion} = N_{He} \cdot A_b \cdot dt \tag{2}$$

where $N_{He}$ is the lifting gas flux and $A_b$ is the effective cross-sectional area of heat and mass transfer.

$$N_{He} = D_{real} \cdot \frac{P_{gas}}{P_{air}} \cdot \frac{(C_{in} - C_{out})}{d_{env}} \tag{3}$$

where $C_{in}$ and $C_{out}$ are the lifting gas and external atmosphere concentrations inside and outside of the aerostat envelope, kmol/$(m^2 \cdot s)$, respectively. $C_{in}$ and $C_{out}$ can be calculated as:

$$C_{in} = \rho_{gas} / M_{gas} \tag{4}$$

$$C_{out} = \rho_{air} / M_{air} \tag{5}$$

where $P_{gas}$ and $P_{air}$ are the pressures of the lifting gas and external air, respectively; $\rho_{gas}$ is the lifting gas density; $\rho_{air}$ is the density of external air; $M_{gas}$ is the molecular weights of lifting gas; $M_{air}$ is the molecular weights of external air; and $d_{env}$ is the thickness of the near-space aerostat envelope.

The transfer velocity is critically affected by the real effective diffusivity. $D_{real}$ is the real effective diffusivity, and it can be calculated as [24]:

$$\frac{1}{D_{real}} = \frac{1}{D_{gas\_air\_eff}} + \frac{1}{D_{K\_gas\_eff}} \tag{6}$$

The effective diffusion coefficient can be calculated by considering the molecular and Knudsen diffusions [25]. $D_{gas\_air\_eff}$ and $D_{K\_gas\_eff}$ are the effective molecular diffusivity and effective Knudsen diffusivity, respectively.

Due to the envelope being made of porous material, $D_{gas\_air\_eff}$ can be calculated by considering the gas diffusion through the porous solid material.

$$D_{gas\_air\_eff} = \frac{\varepsilon_{env}}{\tau_{ss}} \cdot D_{gas\_air} \tag{7}$$

where $\varepsilon_{env}$ is the porosity of the envelope material and $\tau_{ss}$ is the tortuosity. Tortuosity is used in an attempt to account for the longer distance traversed in the pores. Tortuosities usually range between two and six, averaging about three [26].

$D_{gas\_air}$ is the molecular diffusivity, which can be calculated from the Chapman–Enskog equation [27]:

$$D_{gas\_air} = \frac{\alpha_1 \cdot T_{env}^{3/2}}{P_{gas} \cdot r_{col}^2 \cdot \Omega_{gas\_air}} \cdot \left( \frac{M_{gas} + M_{air}}{M_{gas} \cdot M_{air}} \right)^{1/2} \tag{8}$$

where $\alpha_1 = 1.8583 \times 10^{-7}$ [28], $T_{env}$ is the envelope temperature, $P_{gas}$ is the pressure in atmospheres, $r_{col}$ is the collision diameter, and $r_{gas}$ and $r_{air}$ are the collision diameters of the lifting gas and ambient air, respectively.

$$r_{col} = \frac{1}{2} \left( r_{gas} + r_{air} \right) \tag{9}$$

The dimensionless quantity $\Omega_{gas\_air}$ is more complex but usually of an order of one [26]. The detailed calculation is affected by the interaction between the two species [29].

The effective Knudsen diffusivity $D_{K\_gas\_eff}$ can be calculated as:

$$D_{K\_gas\_eff} = \frac{\varepsilon_{env}}{\tau_{ss}} \cdot D_{K\_gas} \tag{10}$$

For a porous envelope material, the Knudsen diffusion is the main influencing factor [30].

$$D_{K\_gas} = \frac{2}{3} \cdot r_P \cdot \sqrt{\frac{8 \cdot R_u \cdot T_{env}}{\pi \cdot M_{gas}}} \tag{11}$$

where $R_u$ is the universal gas constant (8314.47 $\text{Pa} \cdot \text{m}^3 / (\text{kmol} \cdot \text{K})$) [31] and $r_P$ is the pore radius, which is in the order of $10^{-6}$ m.

Based on Equations (7) and (10), we can obtain the porosity of the envelope material and tortuosity, as the air tightness parameters of the envelope material directly affect the effective diffusion coefficient. Analyzing the effective diffusion coefficient in the mass transfer equation as a function of the ratio of porosity to tortuosity, we can obtain the intermediate variable used to indicate the air tightness of materials.

$$ratio = \varepsilon_{env} / \tau_{ss} \tag{12}$$

### 2.2. Balloon Geometry

The expansion of the balloon as a function of altitude can safely be assumed to be governed by the ideal gas law. According to the ideal gas law, the state of an amount of gas can be determined by its pressure, volume, and temperature using the equation:

$$V_b = \begin{cases} V_{gas} & V_{gas} \leq V_{\max} \\ V_{\max} & V_{gas} > V_{\max} \end{cases} \tag{13}$$

where $V_b$ is the volume of the balloon in the current flight environment, $V_{gas}$ is the volume of lifting gas after free expansion in the current flight environment, and $V_{\max}$ is the maximum design volume of the balloon.

$$V_{gas} = \frac{m_{gas} \cdot R_{gas} \cdot T_{gas}}{P_{gas}} \tag{14}$$

where $m_{gas}$ is the mass of the lifting gas, $T_{gas}$ is the thermodynamic temperature of the lifting gas, $R_{gas}$ is the specific gas constant of the lifting gas, $R_{He} = 2078.5 \, \text{J}/(\text{kg} \cdot \text{K})$ for helium, and $P_{gas}$ is the pressure of the lifting gas.

$$P_{gas} = \begin{cases} P_{air} & V_{He} \leq V_{\max} \\ \frac{m_{gas} \cdot R_{gas} \cdot T_{gas}}{V_b} & V_{He} > V_{\max} \end{cases} \tag{15}$$

The differential pressure between the lifting gas and external air, $\Delta P$, can be written as:

$$\Delta P = P_{gas} - P_{air} \tag{16}$$

The balloon envelope thickness (here assuming that the thickness of the skin is uniform and that the thickness is the same everywhere) can be calculated with envelope mass, $m_{env}$, and envelope material, $\rho_{env}$.

$$d_{env} = \frac{m_{env}}{A_b \cdot \rho_{env}} \tag{17}$$

The balloon diameter (top view) can be calculated according to the volume of the balloon. For a high-altitude balloon, $\beta_{dia}$ is a coefficient representing the effect of the balloon volume and shape change on the equivalent diameter of the balloon [32].

$$d_{top} = \beta_{dia} \cdot V_b^{1/3} \tag{18}$$

### 2.3. Dynamic Model

The vertical acceleration of a near-space aerostat is mainly affected by the pressure and temperature of the outside atmosphere. In the vertical direction, the main forces on the aerostat are buoyancy and drag, which will affect the speed change. The following equation demonstrates the influence of these forces on the climbing rate:

$$a_z = \frac{d^2 z}{dt^2} = F_z / m_{tot} \tag{19}$$

where $m_{tot}$ is the total mass of the aerostat system and $F_z$ is the resultant force on the aerostat in the vertical direction, which can be written as:

$$F_z = B - m_b \cdot g - D_b \cdot \left| \frac{v_{z\_rel}}{v_b} \right| \tag{20}$$
$$m_{total} = m_b + m_{add}$$

where $m_{add}$ is the air mass displaced by the motion of the balloon, named the virtual mass, which can be calculated using the virtual mass coefficient $c_{add}$. The virtual mass coefficient

$c_{add}$ can range from 0.25 to 0.5 and is specified as 0.5 here [33]. A precise calculation of the virtual mass can be obtained from references [34,35].

$$m_{add} = c_{add} \cdot \rho_{air}(z) \cdot V_b(z) \tag{21}$$

where $B$ is the total buoyancy of an aerostat.

$$B = V_b(z) \cdot \left( \rho_{air}(z) - \rho_{gas} \right) \cdot g \tag{22}$$

$V_b(z)$ is the aerostat volume.

$$\rho_{gas}(z) = \frac{m_{gas}}{V_b(z)} \tag{23}$$

$F_D$ is the aerodynamic drag, which can be expressed as:

$$F_D = \frac{1}{2} \cdot \rho_{air}(z) \cdot v_b^2 \cdot C_d \cdot A_{eff} \tag{24}$$

$A_{eff}$ is the effective area that used to calculate the aerodynamic drag.

$$A_{eff} = \pi \cdot d_{top}^2 / 4 \tag{25}$$

The influence of the balloon shape is mainly considered in the calculation of the aerodynamic drag coefficient, $C_d$.

$$C_d = \frac{4}{3} \cdot \frac{g \cdot d_{top}}{v_{z\_rel}^2} \cdot \left( 1 - \frac{M_{gas} \cdot T_{air}}{M_{air} \cdot T_{gas}} \cdot \left( 1 + \frac{m_b}{m_{gas}} + \frac{m_{total}}{m_{gas}} \cdot \frac{a_z}{g} \right) \right) \tag{26}$$

### 2.4. Thermal Model

The thermal environment of a near-space aerostat is complicated and consists of direct solar radiation, internal and external infrared radiation, scattered radiation, and internal and external convection [36], as depicted in Figure 4.

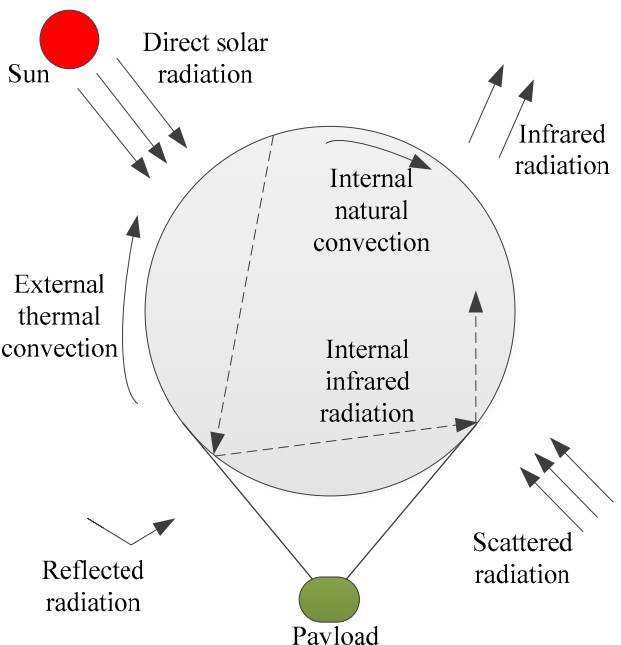

**Figure 4.** Thermal balance of a near-space aerostat.

In this paper, the Earth standard atmosphere was used to analyze the thermal characteristics of the aerostat [37].

$$T_{atm} = \begin{cases} 288.15 - 6.5 \cdot h, \ K & 0 \leq h \leq 11\text{km} \\ 216.15, \ K & 11 \leq h \leq 20\text{km} \\ 216.15 + (h - 20), \ K & 20 \leq h \leq 32\text{km} \end{cases} \tag{27}$$

The atmosphere dynamic viscosity can be expressed by Sutherland's formula:

$$u_a = u_0 \cdot \left( \frac{T_0 + C_a}{T_a + C_a} \right) \cdot \left( \frac{T_a}{T_0} \right)^{3/2} \tag{28}$$

The atmospheric pressure at any altitude can be calculated using the following formula:

$$P_a = P_0 \cdot \left( 1 - \frac{L \cdot h}{T_0} \right)^{M_{air} \cdot g / (R \cdot L)} \tag{29}$$

The change in the lifting gas temperature in the aerostat is mainly related to the convective heat transfer between the gas and the envelope, as well as the volume change of lifting gas. The lifting gas temperature rate can be obtained through:

$$\frac{dT_{gas}}{dt} = \frac{q_{in,gas}}{M_{gas} \cdot c_v} + (\gamma - 1) \cdot T_{gas} \cdot \left( \frac{1}{m_{gas}} \cdot \frac{dm_{gas}}{dt} - \frac{1}{V_{gas}} \cdot \frac{dV_{gas}}{dt} \right) \tag{30}$$

where $\gamma = c_p / c_v$ and $\gamma - 1 = R_{gas} / c_v$. $c_v$ is the specific heat at a constant volume of the lifting gas, $R_{gas}$ is the specific gas constant, and $q_{in,gas}$ is the internal thermal convective heat load [38].

The mass change rate can be expressed as:

$$\frac{dm_{gas}}{dt} = \Delta m_{valve} + \Delta m_{diffusion} \tag{31}$$

The mass change of the lifting gas mainly includes two parts: discharge through the valve, $\Delta m_{valve}$, and gas diffusion through the envelope, $\Delta m_{diffusion}$. The gas diffusion through the envelope can be calculated by the lifting gas diffusion model established in Section 2.1:

$$\Delta m_{valve} = -A_{valve} \cdot c_{disch} \cdot \sqrt{2 \cdot \Delta P_{valve} \cdot \rho_{gas}} \tag{32}$$

$$\Delta m_{diffusion} = -M_{gas} \cdot D_{real} \cdot \frac{P_{gas}}{P_{air}} \cdot \frac{(C_{in} - C_{out})}{d_{env}} \cdot A_b \tag{33}$$

where $A_{value}$ is the outlet area of the valve, $c_{disch}$ is the discharge coefficient, and $\Delta P_{valve}$ is the differential pressure across the area interface.

The envelope temperature differential equation can be expressed as:

$$m_{env} \cdot c_{env} \cdot \frac{dT_{env}}{dt} = q_{in, \ env} \tag{34}$$

The heat flux terms $q_{in,env}$ in Equation (34) are given as:

$$q_{in, \ env} = q_D + q_S + q_{albedo} + q_{IR} + q_{conv} \tag{35}$$

where $q_D$ is the absorbed direct sunlight heat. The absorbed direct solar radiation includes the heat absorbed by the outer surface of the envelope and the heat absorbed by the inner surface through the envelope [39]. The effective absorption of the envelope will be improved due to the multiple types of reflection taking place.

$$q_D = \alpha_{env} \cdot A_{proj} \cdot I_{sun} \cdot \tau_{atm} \cdot \left( 1 + \frac{\tau_{env}}{1 - \gamma_{env}} \right) \tag{36}$$

where $\alpha_{env}$ is the absorption coefficient of the envelope material for solar radiation, $A_{proj}$ is the projected area of the balloon, $I_{sun}$ is the direct solar irradiance, $\tau_{atm}$ is the transmission coefficient of the atmosphere for sunlight, and $\tau_{env}$ and $\gamma_{env}$ are the transmittance and reflectance of the envelope material, respectively [20].

$q_S$ the absorbed scattered radiation and can be expressed as:

$$q_S = \alpha_{env} \cdot \kappa_S \cdot A_b \cdot I_{sun} \cdot \tau_{atm} \cdot (1 + \tau_{env} \cdot (1 + \gamma_{env})) \tag{37}$$

where $\kappa_S$ is the atmospheric scattering empirical coefficient.

$q_{albedo}$ is the ground albedo radiation, and its intensity is mainly affected by the direct solar radiation intensity and the average albedo of the ground.

$$q_{albedo} = \alpha_{env} \cdot A_b \cdot V_F \cdot I_{albedo} \cdot (1 + \tau_{env} \cdot (1 + \gamma_{env})) \tag{38}$$

where $I_{albedo} = \varepsilon_G \cdot I_{sun} \cdot \sin(\alpha_{ele})$, $\varepsilon_G$ is the ground albedo, $\alpha_{ele}$ is the sun elevation angle, and $V_F$ is the angle coefficient between the balloon surface and the Earth's surface, which is related to the flight altitude [40].

$q_{IR}$ is the absorbed infrared radiation heat, which mainly includes ground infrared radiation, $q_{IR,E}$; infrared radiation from the sky, $q_{IR,sky}$; and infrared radiation self-glow from the interior, $q_{IR,env}$.

$$q_{IR} = q_{IR,E} + q_{IR,sky} + q_{IR,env} \tag{39}$$

$$q_{IR,E} = \alpha_{IR} \cdot A_{surf} \cdot I_{IR,E} \cdot V_F \cdot (1 + \tau_{IR}/(1 - r_{IR}))$$
$$q_{IR,sky} = \alpha_{IR} \cdot A_{surf} \cdot I_{IR,sky} \cdot (1 - V_F) \cdot (1 + \tau_{IR}/(1 - r_{IR})) \tag{40}$$
$$q_{IR,env} = \alpha_{env} \cdot \varepsilon \cdot A_{surf} \cdot T_{env}^4 \cdot (1 + \alpha_{IR}/(1 - r_{IR}) + \tau_{IR}/(1 - r_{IR}))$$

$I_{IR,sky} = \varepsilon_g \cdot \sigma \cdot T_{sky}^4 \cdot \tau_{atm\_IR}$, and $T_{sky}$ is the effective temperature of the sky [41]. $q_{conv}$ is the convective heat load on the envelope.

$$q_{conv} = h_{ex} \cdot A_b \cdot (T_{env} - T_{atm}) + h_{in} \cdot A_b \cdot (T_{env} - T_{gas}) \tag{41}$$

The external thermal convection coefficient (which is mainly composed of natural convection, $h_{free-ex}$, and the forced convection heat transfer coefficient, $h_{forced-ex}$) is given as [42]:

$$h_{ex} = (h_{free-ex}^3 + h_{forced-ex}^3)^{1/3} \tag{42}$$

## 3. Model Validation

### 3.1. Verification of the Dynamic Model

Based on the thermal and dynamic models established in this paper, a computer program was developed. The high-altitude experimental data obtained by Yang et al. [33] were used to verify the validity and accuracy of the dynamic model developed in this study. The same balloon parameters, launch times, and launch sites were used in our comparative analysis. These critical parameters are listed in Table 1. During ascent, the differential pressure between the lifting gas and external air was approximately equal to 0 until the maximum design altitude was reached. By comparing the results of the numerical simulation and the experimental data (as shown in Figure 5), it can be seen that the maximum relative error of the flight altitude was less than 5%. There were some differences in flight altitude during ascent, mainly due to the influence of the aerodynamic drag coefficient on the shape change. The variation in balloon flight altitude with the flight time distribution remained almost identical. Hence, the dynamic model developed in this paper can be applied to indicate flight status with reasonable accuracy.

After reaching the design altitude, the pressure and temperature of the internal lifting gas of the aerostat rose rapidly. Then, there were some valve-switching operations that have not been described in the literature; these operations affected the accuracy of temperature prediction based on Equation (30). As can be seen in Figure 6, the prediction result and

reference values corresponded with deviations below 9%, which means that the present thermal model can be used to predict the temperature characteristics of aerostats.

**Table 1.** Design parameters of the near-space aerostat used for comparison [33].

| Parameters | Value |
|---|---|
| Design altitude (km) | 20 |
| Total mass (kg) | 5700 |
| Helium gas mass (kg) | 900 |
| Envelope mass (kg) | 1200 |
| Maximum volume (m$^3$) | 65,000 |
| Absorption coefficient, $\alpha_{env}$ | 0.3 |
| Transmission coefficient, $\tau_{env}$ | 0.5 |
| Launch time | 1 July at 07:00 |
| Launch site | 110° E, 30° N |

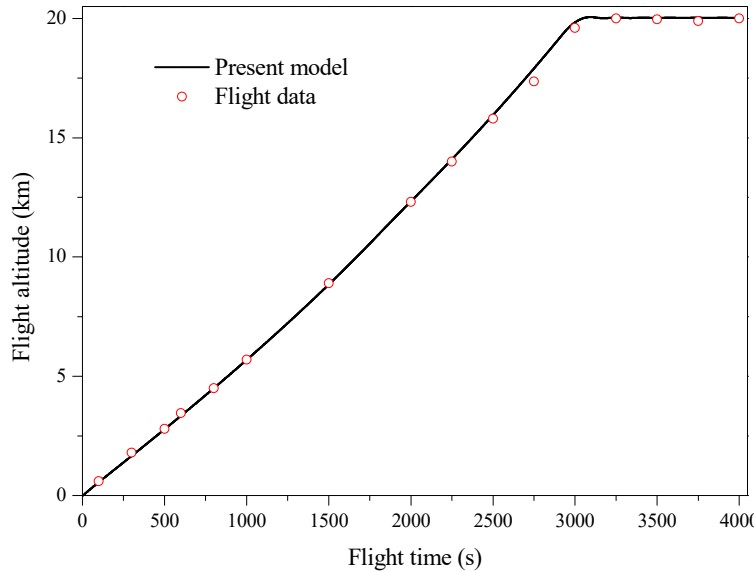

**Figure 5.** Comparison between the predicted data and the experimental data (flight altitude).

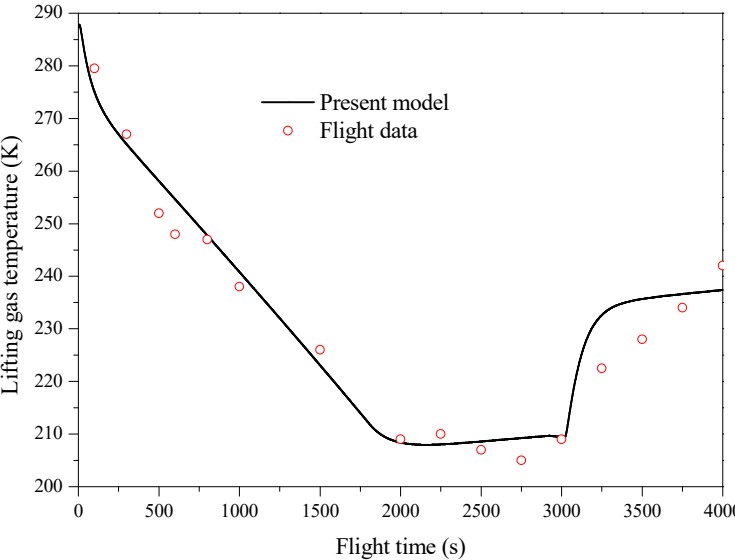

**Figure 6.** Comparison between the predicted data and the experimental data (lifting gas temperature).

### 3.2. Verification of Lifting Gas Diffusion Model

In order to understand the mechanism of the helium leakage of aerostat envelope materials, Wu [43] carried out theoretical and experimental studies. The results of these studies were used to verify the accuracy of our mass transfer model. The design parameters used for analyzing the envelope diffusion coefficient are listed in Table 2. During the flight, the temperature of the balloon envelope generally varied from 216 to 336 K. Therefore, in [43], the diffusion coefficient of an envelope under a range of different envelope temperatures was analyzed. As shown in Figure 7, the diffusion coefficients of envelopes given in the literature and predicted in this paper increased with increases in envelope temperature. In addition, the deviations between the comparison data and the predicted results were less than 6%, indicating that the mass transfer model established in this paper can be used to estimate the lifting gas leakage and flight endurance in the preliminary design of an aerostat.

**Table 2.** Design parameters used for analyzing an envelope diffusion coefficient.

| Parameters | Value |
|---|---|
| Envelope thickness (m) | $2.7 \times 10^{-4}$ |
| Pressures of the lifting gas (mPa) | 0.1 |
| Porosity of the envelope material | 0.2% |
| Tortuosity (kg/(m s$^2$)) | 2 |
| Collision diameters of lifting gas | 2.551 |
| Collision diameters of ambient air | 3.711 |
| Pore radius (m) | $0.2 \times 10^{-6}$ |

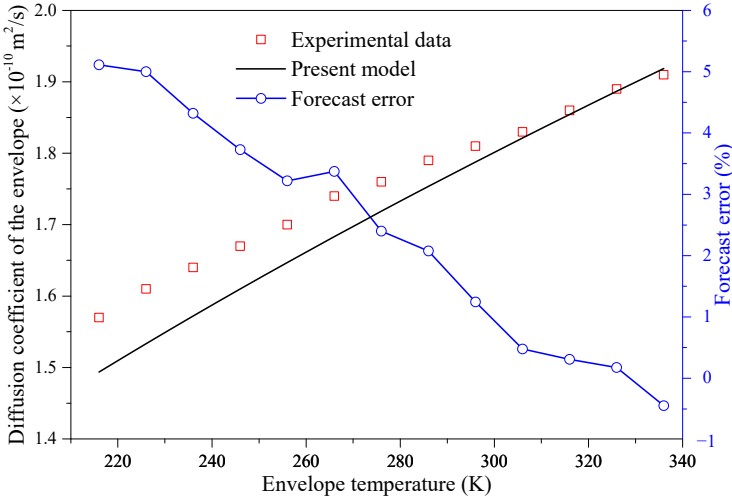

**Figure 7.** Comparisons of the diffusion coefficient of the envelope [43].

## 4. Discussion

### 4.1. Simulation Conditions

The theoretical model established in this paper was used to predict the gas diffusion characteristics and flight performance of a near-space super-pressure balloon. The main design parameters of the balloon are listed in Table 3.

### 4.2. Effects of $\varepsilon_{env}/\tau_{ss}$

Based on the theory established in this paper, it can be stated that the mass of the lifting gas inside a near-space aerostat directly affects the volume and pressure of the lifting gas. Additionally, the total buoyancy of a near-space aerostat varies with the volume of the aerostat at a given altitude. Furthermore, the atmospheric pressure decreases with increases in the altitude, as does the density of the atmosphere. Based on this analysis, it can be summarized that a change in the mass of the lifting gas will affect the flight

altitude of an aerostat. In order to meet the needs of long-term monitoring and long-distance communication, the near-space aerostat—a brand-new concept intended to meet the requirements of long-endurance station-keeping performance—was developed. This means that the flight altitude of an aerostat needs to be guaranteed above a certain limit. Therefore, the effects of the air tightness parameters of the envelope material and the ratio of porosity to tortuosity on the mass of lifting gas inside a near-space aerostat should be studied in detail. For this purpose, five simulations were conducted to study the ratio of porosity to tortuosity ($\varepsilon_{env}/\tau_{ss} = 0.001$, 0.002, 0.004, 0.008, and 0.012).

**Table 3.** Main parameters of the aerostat used for the simulation.

| Parameters | Value |
| --- | --- |
| Design altitude (km) | 20 |
| Failure altitude (km) | 15 |
| Total mass (kg) | 145 |
| Initial helium gas mass (kg) | 20 |
| Payload mass (kg) | 67.2 |
| Envelope mass (kg) | 57.7 |
| Maximum volume (m$^3$) | 1625 |
| Maximum overpressure (Pa) | 600 |
| Launch date | 1 January |
| Launch site | Changsha, Hunan |

Figure 8 presents the variations in the mass of lifting gas inside a near-space aerostat with the $\varepsilon_{env}/\tau_{ss}$ and the ratio of porosity to tortuosity at different flight times. In order to meet the needs of long-term monitoring and long-distance communication, a minimum flight altitude of 15 km was used in this study. In order to achieve station-keeping at this altitude, a sufficient level of lifting gas inside the near-space aerostat had to be maintained. It was found that 16 kg of lifting gas is required for an aerostat to fly at the altitude of 15 km. Under the same flight endurance, the mass of lifting gas inside a near-space aerostat was found to increase with decreases in the ratio of porosity to tortuosity when all aerostats are flying above the lowest altitude. The calculations showed that the amount of lifting gas diffusion taking place increases as the ratio of porosity to tortuosity increases. Additionally, the airtightness of the envelope material tends to worsen over time.

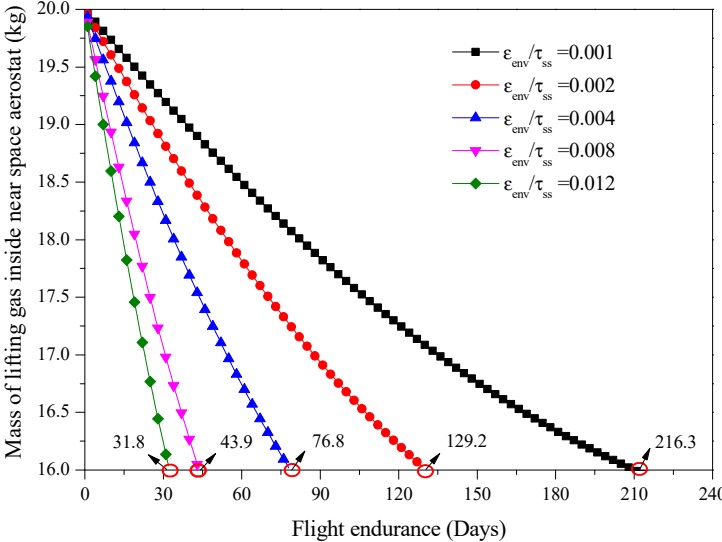

**Figure 8.** Effects of $\varepsilon_{env}/\tau_{ss}$ on the mass of lifting gas inside a near-space aerostat.

Furthermore, it is obvious that flight endurance will increase with decreases in $\varepsilon_{env}/\tau_{ss}$. The flight endurance corresponding to different $\varepsilon_{env}/\tau_{ss}$ values was found to be 31.8, 43.9,

76.8, 129.2, and 216.3 days. It can be concluded that flight endurance at a higher ratio of porosity to tortuosity, $\varepsilon_{env}/\tau_{ss}$, dramatically decreases, whereas it is fairly stable for lower ratios of $\varepsilon_{env}/\tau_{ss}$. This indicates that an optimal ratio of porosity to tortuosity may exist for the selection of envelope materials.

The effects of $\varepsilon_{env}/\tau_{ss}$ on the flight altitude of near-space aerostats are shown in Figure 9. It was found that when the ratio of porosity to tortuosity decreases, the flight altitude increases. Under the same $\varepsilon_{env}/\tau_{ss}$, the flight altitude of a near-space aerostat decreases with the flight time when all aerostats are flying above the lowest altitude. Additionally, the rate of height change gradually increases, especially at smaller $\varepsilon_{env}/\tau_{ss}$.

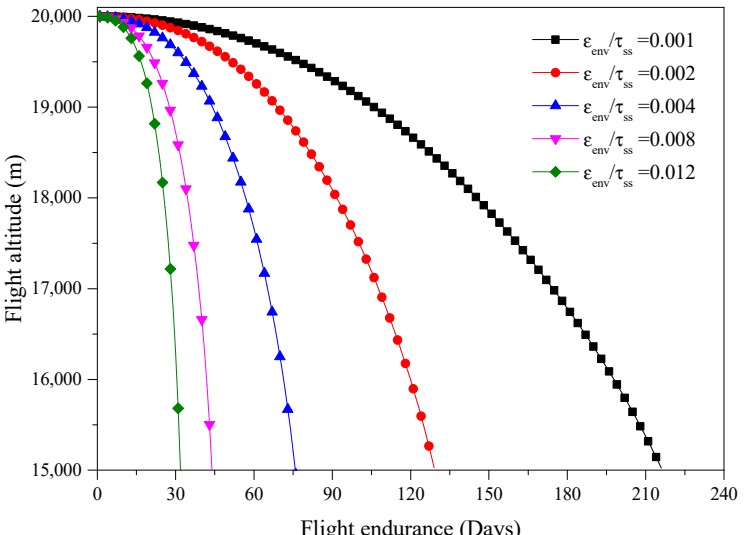

**Figure 9.** Effects of $\varepsilon_{env}/\tau_{ss}$ on the flight altitude of near-space aerostats.

In order to evaluate the effects of $\varepsilon_{env}/\tau_{ss}$ on the diffusion coefficient of an envelope, a series of values for the ratio of porosity to tortuosity were considered. The variation in the diffusion coefficient with $\varepsilon_{env}/\tau_{ss}$ is shown in Figure 10. The results showed that with increases in the flight endurance, the diffusion coefficient decreases. The diffusion coefficient was found to decrease with $\varepsilon_{env}/\tau_{ss}$ due to an increase in the molecular diffusivity. If $\varepsilon_{env}/\tau_{ss}$ becomes larger, the diffusion coefficient decreases faster.

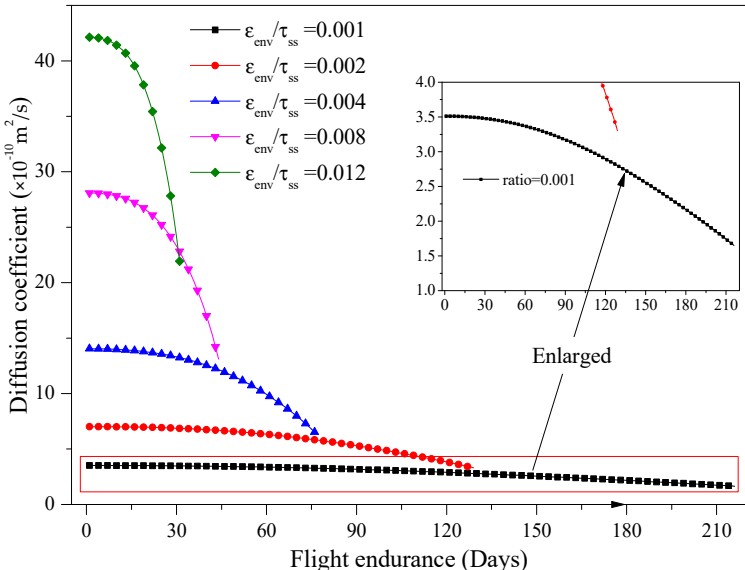

**Figure 10.** Effects of $\varepsilon_{env}/\tau_{ss}$ on the diffusion coefficient of the envelope.

Furthermore, the ratio of $\varepsilon_{env}/\tau_{ss}$ was found to have a significant influence on the envelope and gas temperature, as well as the gas pressure, as shown in Figure 11. With the increase in the flight endurance, the maximum temperatures of the envelope and lifting gas per day were found to first rapidly and then slowly increase before finally decreasing. These results showed that the maximum temperature of the envelope and lifting gas per day are mainly affected by the flight date. The initial flight date was set to be 1 January. With the increase in the flight endurance, the heat caused by direct sunlight was found to gradually increase until the summer solstice. On the same flight date, the higher the $\varepsilon_{env}/\tau_{ss}$ is, the lower the maximum temperatures of the envelope and lifting gas were shown to be, although the difference was small, as shown in Figure 11a,b. This temperature difference is mainly caused by the leakage of lifting gas. The maximum gas pressure per day was found to rapidly increase with the increase in flight endurance, as shown in Figure 11c, because of the decrease in the flight altitude. As shown in Figure 11d, the maximum gas pressure difference ($P_{gas} - P_{air}$) per day was shown to first decrease and then increase with the increase in flight endurance. The main factor influencing the descending section was found to be the leakage of lifting gas, which is caused by the initial high diffusion coefficient of the envelope based on Equation (3). The main factor influencing the rising section was found to be the increase in the lifting gas temperature.

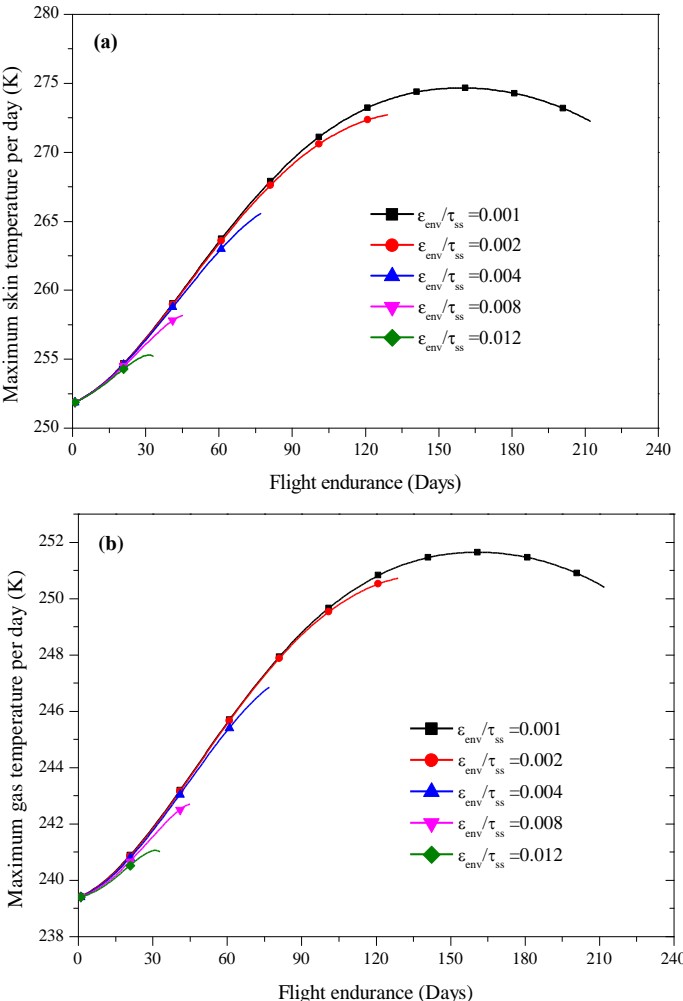

**Figure 11.** *Cont.*

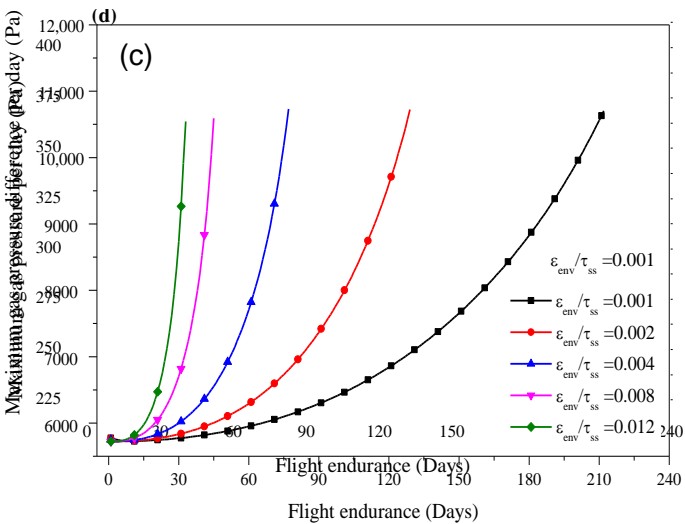

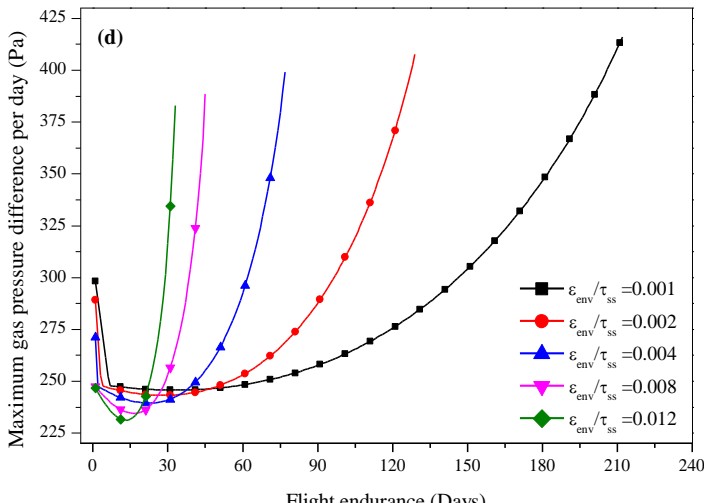

**Figure 11.** Effects of $\varepsilon_{env}/\tau_{ss}$ on the (**a**) maximum skin temperature per day, (**b**) maximum gas temperature per day, (**c**) maximum gas pressure per day, and (**d**) maximum gas pressure difference per day.

### 4.3. Effect of Envelope Radiation Properties

As shown in Figure 12, the effects of the envelope temperature on the lifting gas diffusion rate, envelope thickness, and diffusion coefficient of the near-space aerostat were studied. Under normal conditions, the lowest temperature of a near-space aerostat envelope would be about 210 K, while the highest temperature at noon would be 310 K. Therefore, the lifting gas diffusion rate, envelope thickness, and diffusion coefficient in this temperature variation range were analyzed in this paper. It was found that when envelope temperature rose from 210 to 310 K, the lifting gas diffusion rate increased from 1.33 to 2.32 g/s and the diffusion coefficient increased from $2.75 \times 10^{-10}$ to $4.79 \times 10^{-10}$ m/s$^{-2}$. The envelope thickness of the near-space aerostat decreased with increases in the envelope temperature. The reason for this is that the effective diffusion coefficient, which was calculated considering the Knudsen and molecular diffusions, was found to be positively related to the envelope temperature. This result implies that the envelope temperature can affect the lifting gas diffusion rate by changing the effective diffusion coefficient.

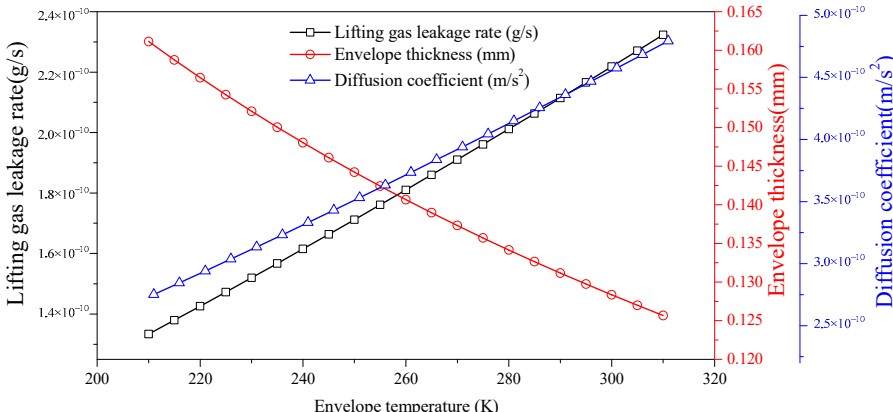

**Figure 12.** The lifting gas diffusion rate, envelope thickness, and diffusion coefficient.

Figure 13 illustrates the temperatures of the lifting gas and envelope and the lifting gas diffusion rate at the summer and winter solstices. It can be seen that there was an obvious peak value of temperature change in the day, which is consistent with the theoretical expectation. The temperatures of the envelope could achieve their highest values, 273.3 and 249.8 K at noon during the summer and winter solstices, respectively. The highest temperatures of the lifting gas were found to be 244.2 and 233.3 K, which were lower than those of the envelope. Furthermore, it is obvious that the temperature differences of the lifting gas and inner air presented the same expressional tendencies. The maximum lifting gas diffusion rate on the day of the summer solstice is obviously larger than that observed on the day of the winter solstice. This result implies that the mass of lifting gas diffused on the day of the summer solstice will be obviously larger than that on the day of the winter solstice.

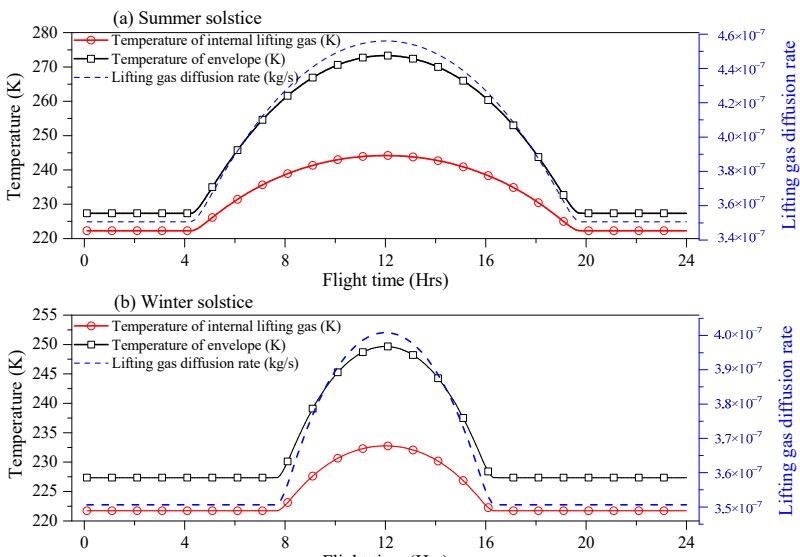

**Figure 13.** The temperatures of the lifting gas and envelope and the lifting gas diffusion rate at the (**a**,**b**).

Examples of how the flight altitude of the near-space aerostat and diffusion coefficient were shown to be influenced by the envelope absorptivity are shown in Figures 14 and 15. Envelope absorptivities of 0.02, 0.05, 0.1, 0.2, and 0.4 were used, as these represent the absorptivity range of most envelope materials. Under the same level of flight endurance, the flight altitude of the near-space aerostat was shown to increases with decreases in the envelope absorptivity when the aerostat was flying above the lowest altitude. It can be

seen in Figure 15 that over the whole flight period, the diffusion coefficient first slowly and then sharply declined as the flight time increased. The results showed that the trends of the flight altitude of the simulated and measured diffusion coefficients were consistent. These results indicate that a higher envelope absorptivity will result in a higher temperature of the lifting gas and envelope, which will consequently increase the mass of lifting gas diffused. Based on Equation (8), the diffusion coefficient is a function of $T_{env}^{3/2}/P_{gas}$. As can be seen from Figure 16, the maximum skin temperature and maximum gas pressure per day were found to gradually increase along with the flight endurance, but the rate of change of rising speed was different, leading to a nonlinear downward trend.

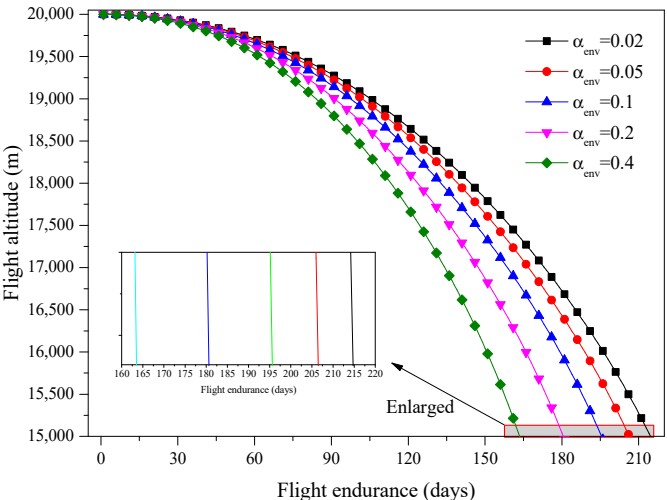

**Figure 14.** Effects of envelope absorptivity on the flight altitude.

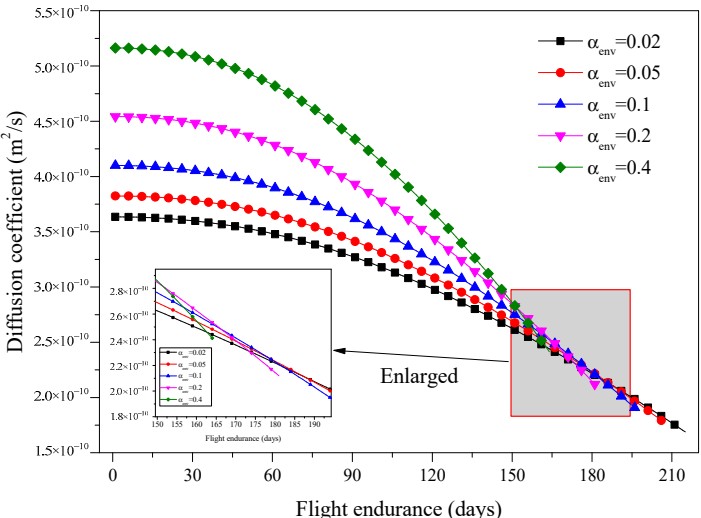

**Figure 15.** Effects of envelope absorptivity on the diffusion coefficient.

The effects of envelope absorptivity on the envelope and gas temperature, as well as the gas pressure, were studied in order to analyze the mechanism of the effect of absorptivity on the diffusion properties, as shown in Figure 16. With the increase in the flight endurance, the maximum temperatures of the envelope and lifting gas per day firstly increased and then decreased, as shown in Figure 16a,b. On the same flight date, the envelope temperature with a higher absorptivity was significantly higher than that with a lower absorptivity. The maximum gas pressure and maximum gas pressure difference per day increased with the increase in envelope absorptivity, as shown in Figure 16c,d.

As shown in Figure 17, five envelope emissivities were considered here to investigate their impact on the flight altitude of a near-space aerostat. Several common envelope

materials were considered for this parametric study. It can be seen that the flight altitude of the aerostat decreased as the flight time increased. The flight endurances of the near-space aerostat under these situations remained fairly steady for 190 days. The results of this research demonstrate that the envelope emissivity has some effect on the flight altitude. Figure 18 presents the effects of the envelope emissivity on the diffusion coefficient. It can be easily seen that the diffusion coefficient was reduced by almost 50% as the flight time increased. The effects of the envelope emissivity on the envelope and gas temperature, as well as the gas pressure, were studied in order to analyze the mechanism of the effect of the envelope emissivity on the diffusion properties, as shown in Figure 19. With the increase in the flight endurance, the maximum temperatures of the envelope and lifting gas per day firstly increased and then decreased, as shown in Figure 19a,b. On the same flight date, the temperature of the envelope with a lower emissivity was significantly higher than that of the envelope with a higher emissivity. The maximum gas pressure and maximum gas pressure difference per day decreased with increases in the envelope emissivity, as shown in Figure 19c,d.

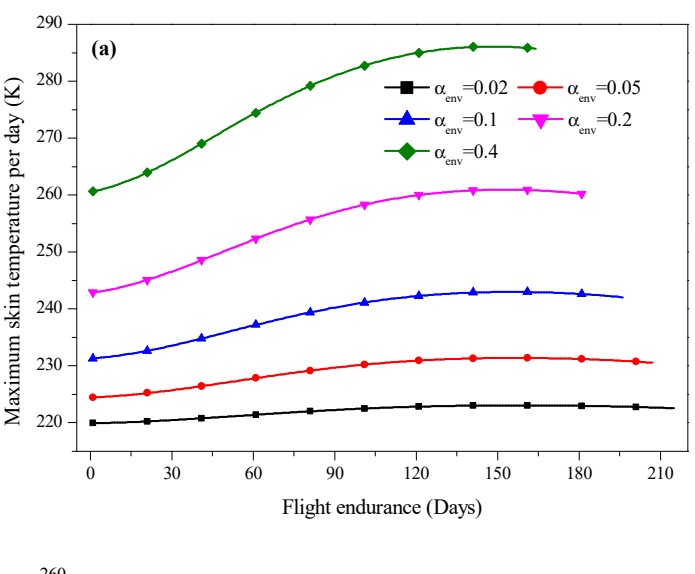

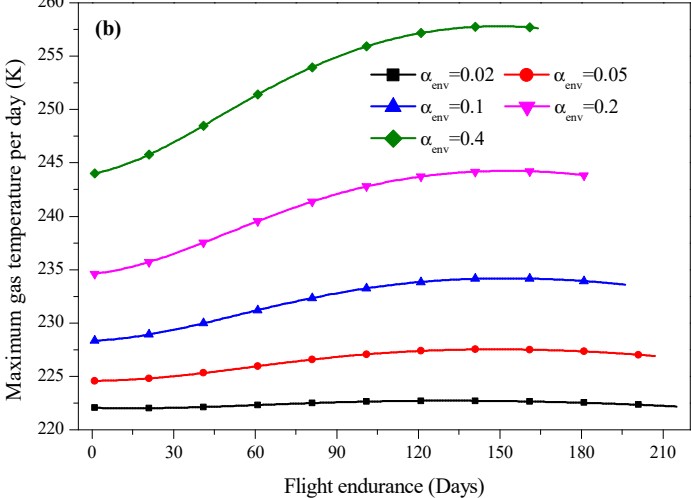

**Figure 16.** *Cont.*

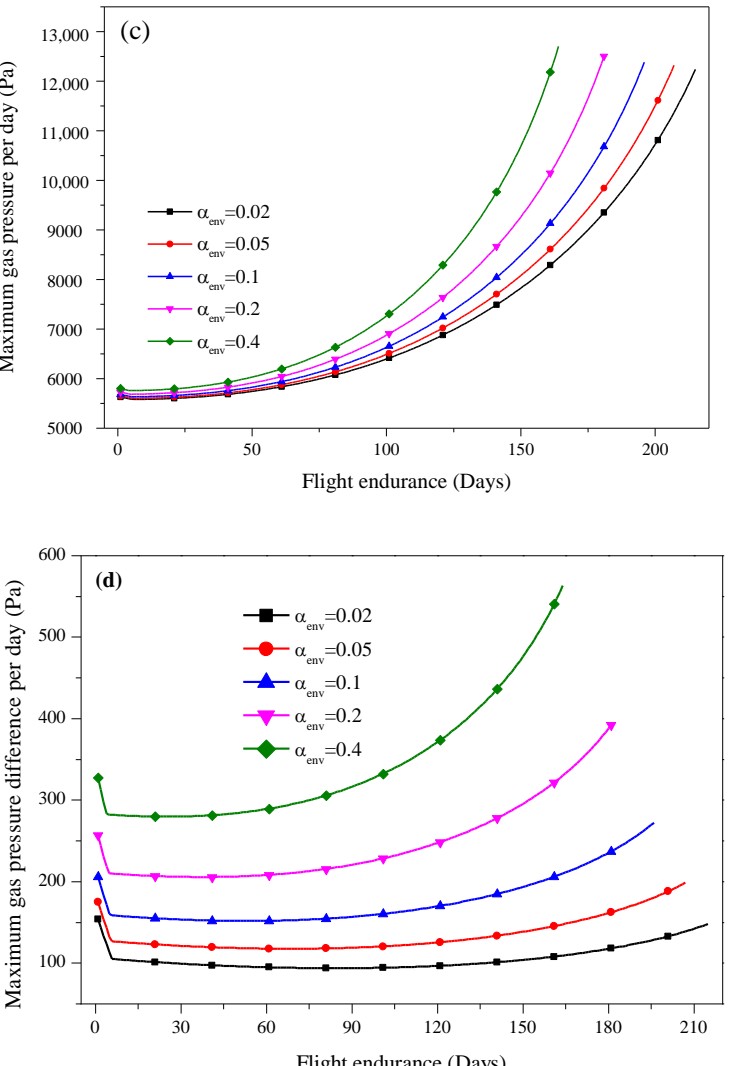

**Figure 16.** Effects of envelope absorptivity on the (**a**) maximum skin temperature per day, (**b**) maximum gas temperature per day, (**c**) maximum gas pressure per day, and (**d**) maximum gas pressure difference per day.

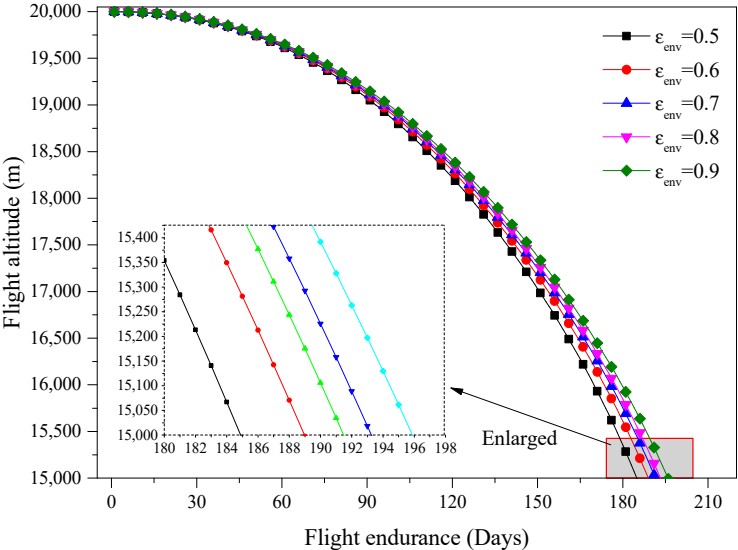

**Figure 17.** Effects of envelope emissivity on flight altitude.

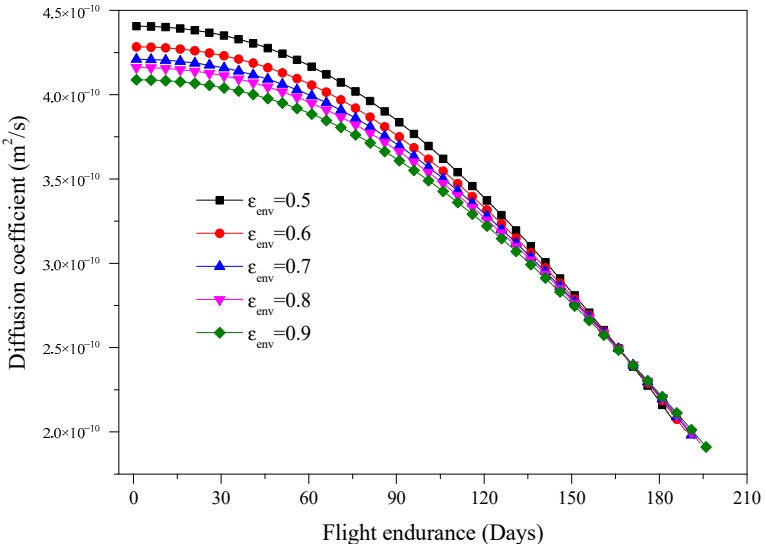

**Figure 18.** Effects of envelope emissivity on diffusion coefficient.

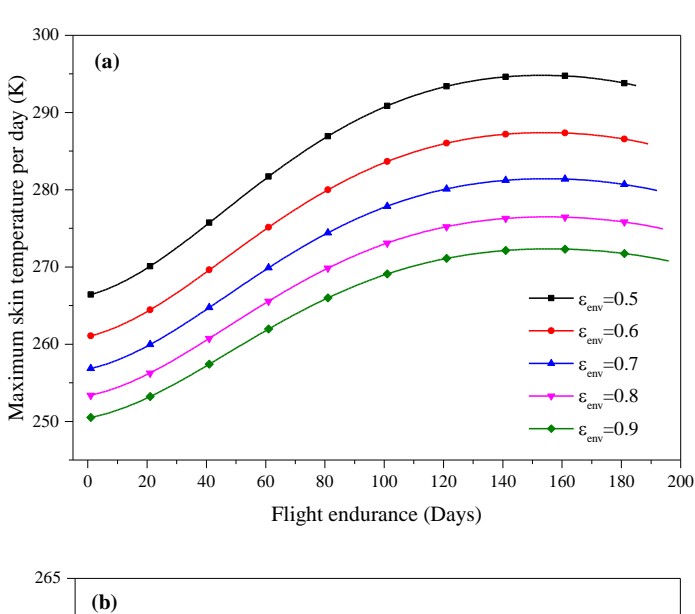

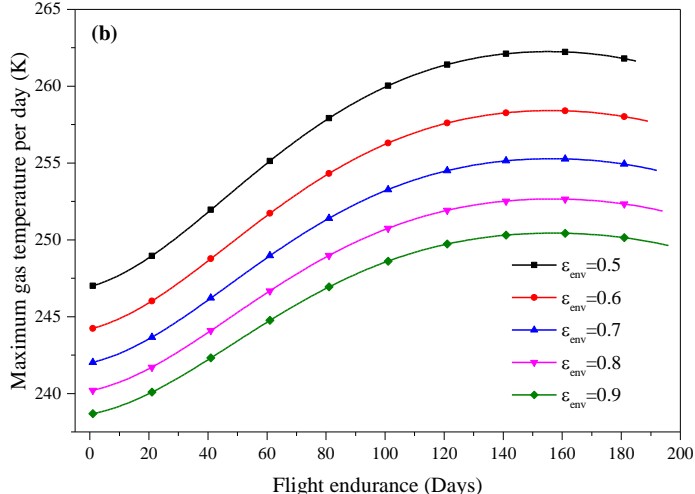

**Figure 19.** *Cont.*

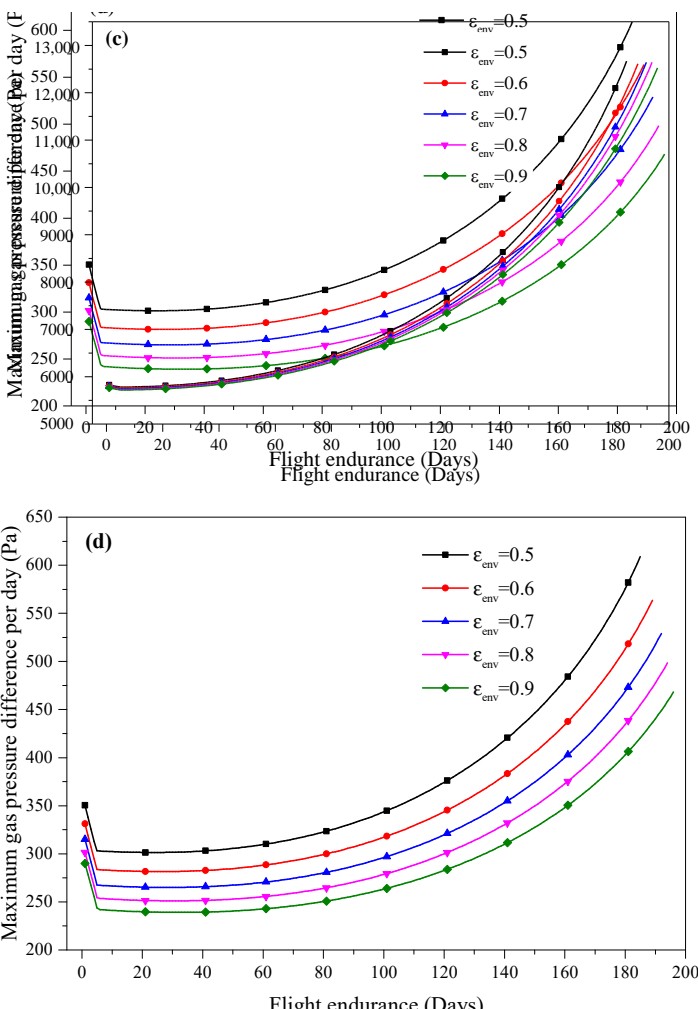

**Figure 19.** Effects of envelope emissivity on the (**a**) maximum skin temperature per day, (**b**) maximum gas temperature per day, (**c**) maximum gas pressure per day, and (**d**) maximum gas pressure difference per day.

## 5. Conclusions

Based on the lifting gas diffusion model and dynamic model with thermal effect established in this paper, the lifting gas diffusion behavior and the effects of the envelope properties on the flight performance of the aerostat were numerically investigated and validated. The main results were as follows:

(1) Based on the verification of results, it was found that the lifting gas diffusion model proposed in this paper, which considers the thermal effect for a near-space aerostat during a long-endurance flight, can be utilized to study lifting gas permeability and flight performance.

(2) The ratio of porosity to tortuosity was found to have a significant influence on the gas diffusion coefficient, directly leading to a sharp decline in flight endurance with increases in the ratio of porosity to tortuosity. In the preliminary design of an aerostat, it is helpful to choose an envelope material with an optimal ratio of porosity to tortuosity in order to improve the aerostat's flight performance.

(3) During high-altitude flight, the lifting gas diffusion rate and diffusion coefficient are very sensitive to changes in envelope temperature. Compared to the envelope infrared emissivity, the envelope absorptivity was found to have a stronger influence on the lifting gas diffusion and the thermal performance of near-space aerostats. A higher envelope absorptivity would result in higher temperatures of the lifting gas and envelope, which would consequently increase the mass of lifting gas diffused.

**Author Contributions:** Conceptualization, J.L. (Jun Li) and L.L.; methodology, Z.C. and L.L.; software, J.L. (Jun Liao); validation, J.L. (Jun Liao), J.L. (Jun Li) and L.L.; investigation, S.L.; data curation, L.L.; writing—original draft preparation, L.L. and Z.C.; writing—review and editing, J.L. (Jun Liao) and J.L. (Jun Li); visualization, J.L. (Jun Li); supervision, S.L. All authors have read and agreed to the published version of the manuscript.

**Funding:** This work was supported by the Key R&D Projects of Hunan Province (2021GK2011) and the Natural Science Foundation of Hunan Province No. 2019JJ50773, as well as National Key Research and Development Program of China (2016YFB0500801). This research was also supported by the Innovation-Driven Project of Central South University (No. 2018CX024).

**Institutional Review Board Statement:** Not applicable.

**Informed Consent Statement:** Not applicable.

**Data Availability Statement:** Not applicable.

**Conflicts of Interest:** The authors declare no conflict of interest.

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
