# Peer review of "Effect of Lifting Gas Diffusion on the Station-Keeping Performance of a Near-Space Aerostat"

_aerospace, doi:10.3390/aerospace9060328_

Round 1
Reviewer 1 Report
The authors addressed the reviewer's questions and clarification/correction requests in a very transparent, substantial and fully satisfactory way. The revised version is a significant improvement compared to the initial submission. The reviewer recommends to accept the manuscript in the present form.
Author Response
Thank you very much for your comments and suggestions on our manuscripts. Your comments are very helpful for revising and improving our manuscript.
Reviewer 2 Report
The paper has been substantially improved allowing the reader to fully understand the novelties proposed by the authors.
I just have to point out that the authors must specify that the equation (21) for the calculation of the added masses is a very simplified formula and that for a precise calculation of these masses the reader can consult the references below:
Korotkin, A., (2009), “Added masses of ship structures”. Springer-Verlag. New York.
S. Chaabani, N. Azouz, “Estimation of the Virtual Masses of a Large Unconventional Airship based on Purely Analytical Method to aid in the Preliminary Design”. Aircraft Engineering and Aerospace Technology (2022). Vol. 94, No. 4, pp. 531-540.
Author Response
Thank you very much for your comments and suggestions. Your comments are very helpful for revising and improving our paper.
(1) I just have to point out that the authors must specify that the equation (21) for the calculation of the added masses is a very simplified formula and that for a precise calculation of these masses the reader can consult the references below:
Korotkin, A., (2009), “Added masses of ship structures”. Springer-Verlag. New York.
- Chaabani, N. Azouz, “Estimation of the Virtual Masses of a Large Unconventional Airship based on Purely Analytical Method to aid in the Preliminary Design”. Aircraft Engineering and Aerospace Technology (2022). Vol. 94, No. 4, pp. 531-540.
Responses:
Thank you for bringing this concern to our attention. We have added references in the revised version according to the reviewers' comments.
Reviewer 3 Report
The first point is that the Abstract and the Conclusion seems too long. Further, in the Conclusion you should avoid equations and numerical relations, it is recommended to use just conceptual conclusions.
An extensive editing of ENGLISH is necessary in this paper. There are a lot of English errors, like those 7 sentences below.
1) ...the diffusion model was developed obtain the diffusion coefficients of the composite laminates. 2) A great many of experts and scholars have studied the thermal performance of the aerostat ... 3)...it is very necessary to understand the lifting gas diffusion characteristics. 4) The numerical model is used to analyse the lifting gas diffusion and study the effects of envelope properties on the on the station-keeping endurance of the aerostat in detail. 5) The results obtained from the analysis of the lifting gas diffusion can contributes to improve the flight performance a.... 6) Based on the lifting gas diffusion model and dynamic model with thermal effect established in this paper, the lifting gas diffusion behavior, the effects of envelope properties on the flight performance of the aerostat were numerically investigated and validated. [THE PROBLEM HERE IS THAT THE SENTENCE IS TOO LONG...] 7) It can be believed that the research of coupling effects of the real-time wind data...Author Response
Thank you very much for your comments and suggestions. Your comments are very helpful for revising and improving our paper.
(1) The first point is that the Abstract and the Conclusion seems too long.
Responses:
Thank you for bringing this concern to our attention. We have added references in the revised version according to the reviewers' comments.
The statements of "The results obtained from the analysis of the lifting gas diffusion can lay a solid foundation for improving the flight performance of near-space aerostats, as well as providing improved design considerations for aerostats." was deleted.
In the conclusion, the statements of "Based on the conclusions of this paper, it can be seen that lifting gas diffusion can influence the flight altitude. According to the actual wind field conditions of different regions, the flight trajectory and flight distance of near-space aerostats will have a large effect on changes in the flight altitude. It can be stated that research on the coupling effects of real-time wind data, lifting gas diffusion, and flight latitudes and dates may provide more reliable and accurate information about the flight trajectory and station-keeping performance of near-space aerostats. Therefore, works on these coupling effects should form the basis for further investigations in this area" was deleted.
(2) An extensive editing of ENGLISH is necessary in this paper. There are a lot of English errors, like those 7 sentences below.
Responses: Thank you for bringing this concern to our attention. We have used the English editing service recommended by the editor. And we corrected the English errors in the full text, including the seven sentences you listed.
This manuscript is a resubmission of an earlier submission. The following is a list of the peer review reports and author responses from that submission.
Round 1
Reviewer 1 Report
The authors present the theoretical foundation for a numerical model which describes the dynamics of a high-altitude aerostat. The balloon dynamics includes the modeling elements for the flight dynamics, thermal behavior and lifting gas loss. The latter element is the focus of this manuscript. Validation, verification and simulation results complement the manuscript.
Chapter “Introduction”: The introduction cites various balloon mission concepts and underlying balloon hull types. It should become clearer, which type of balloon and hull layer build-up this work focuses. E.g. is a zero-pressure balloon or super-pressure used? Is the hull in this manuscript assumes to be a multi-layer build-up or a single layer foil? This would be necessary for readers to understand the constraints and applicability of the following theoretical work.
Line 98 to 102 it is not entirely clear if the description refers to the cited work and/or is the baseline for the authors work. Please keep separate the review of the reference work and your own baseline.
Figure 2: Is this an own analysis (imagery) by the authors or taken from other references? Please clarify.
Figure 3: the used graphics was used already in Kayhan & Hastaoglou (your reference 41) and is reprinted here with slight modifications to the text annotations. Please cite properly, such “redrawn from …”.
Line 135/136: English wording / grammar incorrect. Please correct.
Theoretical model, line 165: the theoretical foundation is based on the concept of “porous material”. Can this assumption be applied to any balloon hull material (single or multi-layer) or is this restricted to certain materials (e.g. composites)? Please clarify the applicability, also in the context of the question above regarding the balloon hull type and build-up.
Equation 14 and Line 200: What does the parameter (Greek letter) omega mean and where is its value coming from?
Equation 14: the delta_P term refers to zero-pressure balloons. Please clarify and ensure throughout the manuscript whether you refer to zero- or super-pressure balloon type.
Equation 19: I think “rho air” should read “rho gas”. Why is this multiplied with 0.5? Please correct/clarify.
Equation 22 (aerodynamic drag): the letter D is already introduced in equation 6 as diffusivity. Please consider an unambiguous parameterization.
Thermal model lines 233 to 250, also l. 260 to 287: this is described by various sources already. Please add relevant references.
Equation 29: I think this one of the most important equations in this manuscript. The authors should highlight that the second part of it (the diffusion term) is coming from the earlier part of this manuscript.
Chapter 3 “Validation”: the source and background of the used experimental data must become more traceable to demonstrate model validity. E.g. has the used experimental data a reference, or is this an own investigation? Please give details and insight into what was done and how it compares to the simulated configuration. Again, the type of balloon would be important to know here.
As the theoretical foundation shows, the flight dynamics is strongly coupled to the thermal dynamics of the system. Therefore, it is necessary to not only show the altitude profile but also the thermal evolution. Please add, or at least provide context information on the thermal environment.
The validation data is limited to only about 3 hours of flight although the balloon systems seems to continue to fly. Please explain this limitation. Especially in the context of the important thermal and diffusion model, a full diurnal cycle would be relevant to validate those parts of the model as well.
On the other hand, the experimental data looks very close to simulation data produced by or shown in Kayhan [your ref. 41]. Is this used as simulation benchmark instead? Please clarify.
Discussion, line 328: Ratios of porosity to tear stress: is this an arbitrary choice of values to show the sensitivity of the model to those values or are these values linked to specific types of materials, composites, or other? Please clarify.
Line 332: should read “is given”.
Figure 13: why are the graphs converge towards day 150? Please explain in the text.
Figure 16: this figure implicitly introduces also the horizontal motion of the balloon system. The horizontal flight dynamics should be introduced then as well (or at least with an appropriate reference) and the underlying wind model/assumption shall be mentioned. On the other hand, your manuscript focuses on station keeping performance. In this aspect, the horizontal part of the trajectory might be of less relevance to understand the model. In case it does not add further understanding of the station keeping performance, the authors might want to restrict the paper to only the vertical part of the model.
Reviewer 2 Report
The paper deals with a particular problem of the analysis of carrier gas diffusion in near space airships and its influence on flight endurance, and flight altitude. This problem is rarely addressed in the study of airships. Based on well-established relationships and a basic dynamic model taking thermal effects into account, the authors were able to establish a numerical model and show the influence of the ratio of porosity to shear stress on flight autonomy.
A few questions and remarks:
- Have you dealt with the influence of gas diffusion on the rigidity of the airship? because the flight model of a flexible airship is not identical to that considered rigid.
- Have you planned a regulation of the internal pressure in the hull according to the diffusion of gases?
- The current fashion is the development of pre-design algorithms for airships, in particular the volume, shape and flight altitude of these machines (for example reference 3). Do you think your model could help refine these algorithms?
Remarks
on page 5 the stress unit is wrong.
The authors should list the innovations of this paper one by one in Introduction.
Check all the grammar errors in the text and be careful with the format and layout of the paper, as well as the citation format of the references.
Reviewer 3 Report
The paper addresses the issue of loss of helium from a high-altitude balloon due to membrane permeability. The topic is very interesting for long-duration applications making use of lighter-than-air vehicles and is definitely not treated sufficiently in the literature, so that this study has potential for an original contribution. A set of relations are presented that model the thermal and membrane permeability aspects of the system. Unfortunately, the values of most of the many empirical parameters on which the model is built are not disclosed; the numerical simulations results are presented without any information on the computational implementation of the model, providing no insight on the many inevitable assumptions that such practical implementation requires. Summary data from a single, unspecified flight are presented as a validation of the dynamic model; the permeability model is also declared valid by showing only one summary curve, under completely unknown boundary conditions, constraints and assumptions. Considering all the important infomation omitted, the simulations presented and the conclusions of the work are not credible.
Line 33 - space-based aircraft : maybe space-based vehicles
Ref. 7 is not really about station-keeping performance requirements
Line 38: misleading - it is not about tethered balloons, but rather a dual-balloon or balloon-sail assembly connected by tether
Fig. 1a and Fig. 1b are from Ref. 13. This is not clearly stated in the text, nor is it clear whether permission is given to use such material. The balloon is of a very special type, intended to fly in the upper atmosphere of Venus; this should be pointed out and discussed properly.
Line 81 - “Figure 2 shows the microstructure of the developed film fabric laminate” : was it developed by the authors? Please clarify and, if needed, mention and/or provide reference to the developer.
Line 92 - Please define the crack intersection angle.
Ref. 24 seems to be unrelated to near space aerostats.
Ref. 28 is related to the thermal design of the payload box, not of the aerostat balloon; it seems to be inappropriate in the context of this paper.
Line 128 - typo: “on the on the”
Eq. 25: for clarity, state that Tatm is in K
Section 3.1 - a graph is shown (Fig. 5) and conclusions of supposed general validity are drawn from it; but no information is provided on the specific flight data shown, such as the characteristics of the balloon used, the mass of the system, the quantity of lifting gas, the controller used to stabilize altitude (or lack thereof), the instrumentation used to measure altitude and its accuracy, etc. Similarly, no information is provided on how the relations presented in the previous section have been actually used to build a numerical simulation; what values have been used for the many empirical coefficients in the formulas; etc. Lacking such information, the conclusions are not supported. It is not even clear whether the flight measurements were taken by the authors or are the work of others. In addition, a single set of data from a single flight is not enough to conclude that a numerical model will yield correct results under different initial conditions.
Section 3.2 - to a lesser extent, similar considerations as for Section 3.1 apply; again, no details are given on the numerical model and a single case is presented (Fig. 6). The remainder of the paper presents numerical results the validity of which is very much questionable, given the lack of visibility on the numerical model and the single-point data set used to validate it.
Line 334 - The figure given for the mass of lifting gas refers, necessarily, to some specific configuration of the aerostat (size of the balloon, payload mass, etc.), which is however undisclosed. The procedure used to calculate such amount of gas is also kept out of the readers’ reach.
In Section 4.3 a 3-D trajectory simulation is presented; again, there is no information about the assumptions used in modeling, e.g. the velocity distribution of the wind, the horizontal drag characteristics of the balloon, etc.